# Association between HIV and treatment-resistant hypertension in Malawian adults: a protocol for a case–control study

Josephine Gondwe,[1] Maclean Ndovie,[2] Felix Khuluza,[3] Clifford George Banda  [1]

[1]Population Health Theme, Malawi-Liverpool-Wellcome Research Programme, Blantyre, Malawi
[2]Department of Medicine, Queen Elizabeth Central Hospital, Blantyre, Malawi
[3]Department of Pharmacy, Kamuzu University of Health Sciences, Blantyre, Malawi

**Correspondence to**
Clifford George Banda;
cgbanda@mlw.mw

## ABSTRACT

**Introduction** Treatment-resistant hypertension (RH), defined as uncontrolled blood pressure (≥140/90 mm Hg) despite treatment with ≥3 medications of different classes (including diuretics) at optimal doses, is associated with poor prognosis and an elevated risk of end-organ damage. In areas where HIV is endemic, such as sub-Saharan Africa, the risk of hypertension is high in people living with HIV. It remains unknown if HIV infection further increases the risk of RH. This study seeks to determine the association between HIV and RH as well as investigate other factors associated with RH in hypertensive Malawian adults.

**Methods and analysis** A case–control study will be conducted among adult hypertensive patients attending a clinic at a referral hospital in Malawi. The cases will be hypertensive patients with a confirmed diagnosis of RH. For each case, two controls (hypertensive patients without RH), frequency matched for age group and sex, will be selected from among hospital clients attending the same hypertension clinic as the case. In both groups, HIV status will be ascertained. Additionally, information on other potential risk factors of RH, such as chronic kidney disease, obesity, hypercholesteraemia, diabetes, smoking, alcohol use, antiretroviral therapy regimen and duration, will be collected in both cases and controls. For each of the potential risk factors, ORs will be calculated to quantify the strength of their association with RH. In a multivariate analysis, conditional logistic regression will be used to assess the independent association between HIV and RH as well as the influence of the other potential drivers of RH.

**Ethics and dissemination** This protocol has been approved by the College of Medicine Research Ethics Committee (COMREC) in Malawi (P.05/22/3637). Findings from this study will be disseminated through a peer-reviewed publication in an open-access international journal. Furthermore, anonymised data will be available on request from the authors.

## INTRODUCTION

Over the last 20 years, sub-Saharan Africa (SSA) has faced a rise in the prevalence of non-communicable diseases (NCDs) because of an increase in cardiovascular risk factors such as poor diet, little to no physical activity, hypertension, obesity, diabetes and dyslipidaemia.[1] This rise presents a double burden on SSA since the region is also faced with a high prevalence of infectious diseases like tuberculosis, malaria and HIV. Previous studies have shown a growing intersection between infectious diseases and NCDs, for example, an increased prevalence of hypertension among people living with HIV.[2 3] In this subgroup, several pathophysiological processes have been implicated in the causation of hypertension. These include microbial gut translocation, elevated inflammatory markers like interleukin-6,[4] antiretroviral therapy (ART)-related inflammation due to protease inhibitors, immune reconstitution syndrome, HIV- related renal disease and lipodystrophy; caused by both HIV and ART use.[2 5]

Despite its high global burden, most people who suffer from hypertension, including those living with HIV, fail to achieve adequate blood pressure control and are usually unaware of their disease status, until faced with end-organ disease. In SSA, only 27% of people with hypertension are aware of their disease, 18% are receiving treatment and yet only 7% of those suffering

from hypertension have well-controlled blood pressure.[6] Inadequate blood pressure control may be due to several factors, including non-adherence by physicians to treatment guidelines, pharmacological treatment noncompliance, lack of physical activity or the presence of white-coat hypertension.[7] Treatment-resistant hypertension (RH) is another increasingly recognised cause of uncontrolled blood pressure that occurs in individuals who are adhering to optimised treatment regimens of antihypertensive medications.[3]

### STRENGTHS AND LIMITATIONS OF THIS STUDY

⇒ A key strength of the case–control design is that it will allow the investigation of multiple exposures that could drive treatment-resistant hypertension in people living with HIV.
⇒ To the best of our knowledge, this is the first study in Malawi to understand drivers of treatment-resistant hypertension and will generate important knowledge to inform future cohort studies in settings where HIV is highly prevalent.
⇒ The study will only recruit participants from one large hypertension clinic at a referral public hospital. This has the potential to exclude other adults that could be otherwise eligible (eg, those from private hospitals or inpatients) leading to selection bias.

RH is defined as uncontrolled blood pressure (≥140/90 mm Hg) despite treatment with ≥3 medications of different classes (including diuretics) at optimal doses. RH is associated with poor prognosis and elevated risk of end-organ damage which can lead to premature deaths. The global burden of RH has been estimated to be around 10.3%,[3] and previous studies have estimated the burden of RH in Africa to be ~12%.[8] Additionally, RH is high among obese individuals, older people (particularly>75 years) and those with chronic conditions such as chronic kidney disease, diabetes mellitus, hypercholesteraemia, heart failure and stroke.[7–9] However, given the geographic intersection between HIV and hypertension, there is a paucity of evidence on whether HIV infection increases the risk of RH.

Malawi, like most SSA countries, has a high burden of HIV: 10.6%[10] and a high prevalence of hypertension: ~16%.[11] In this study, we hypothesise that HIV increases the risk of RH in hypertensive Malawian adult patients. Thus, we aim to determine the independent association between HIV infection and RH. Additionally, we will investigate the impact of other potential drivers of RH such as hypercholesteraemia, obesity, chronic kidney disease, diabetes, ART regimen and duration, alcohol use as well as smoking. Understanding whether HIV increases the risk of treatment-RH, and the influence of other potential drivers of RH is vital in targeting early optimal preventive and therapeutic interventions in this vulnerable subpopulation.

## METHODS AND ANALYSIS
### Study design and setting
A case–control study will be conducted at the outpatient hypertension clinic at Queen Elizabeth Central Hospital (QECH) in Blantyre, Malawi. QECH is the largest tertiary hospital in Malawi that receives referrals from across the country. It is a free public hospital hence it serves mainly the low-income to middle-income class. A case–control study design has been chosen for three reasons. First, treatment-RH is a relatively rare disease, and a case–control study will allow research on this outcome. Second, it will reduce the time constraint that a prospective cohort study would present due to the need for a longer follow-up period. Third, it will allow the investigation of multiple factors associated with RH.

### Identifying cases and controls
Cases will be defined as hypertensive patients with a confirmed diagnosis of treatment-RH. In this study, treatment-RH will be defined as a blood pressure of ≥140/90 mm Hg measured using an ambulatory blood pressure monitor (ABPM) over 4 hours in patients with hypertension and who are on at least three different medications including a diuretic. The 4 hour ABPM will be adopted in this study since previous studies have shown and validated a 4 hour ABPM as an accurate proxy of daytime (24 hours) blood pressure.[12] Controls will be defined as hypertensive patients with well-controlled (<140/90 mm Hg) regardless of how many antihypertensive medications they are currently taking. For each case, two controls, frequency matched for age group and sex, will be selected from among hospital clients attending the same hypertension clinic as the case (ie, from the population giving rise to the case). The study period is from January 2022 to August 2023, with recruitment expected to conclude in June 2023. Patients who are 18 years and above presenting to the hypertension clinic at QECH will be considered for recruitment into the study. All patients presenting to the clinic with uncontrolled BP, and on at least three different medications including a diuretic, will be considered as potential cases, and screened for potential recruitment as outlined in figure 1. Controls will be selected, using simple random sampling, from among a cohort of patients presenting to the same clinic, but without RH. The selection of both controls and cases will be done before the ascertainment of HIV status. Additionally, selecting controls from a population that would be giving rise to the cases as well as the use of simple random sampling will minimise any potential selection bias.

### Recruitment
Suspected RH cases will be patients who have elevated blood pressure of ≥140/90 mm Hg on the day of recruitment and who are taking any three classes of antihypertensive medication including a diuretic. The investigators will measure the blood pressure of both cases and controls using a digital blood pressure cuff. After 30 min of the participants' rest and while the participant is seated, the investigators will collect three blood pressure readings 5 min apart. This method was used in previous studies to measure blood pressure in Malawian adults.[11] For participants who have pseudo-RH (defined as elevated blood pressure≥140/90 mm Hg on at least three medications including a diuretic, but where there is evidence of white-coat hypertension, or where the cause of the uncontrolled blood pressure is known),[9] ABMP will be used (as illustrated in figure 1). As part of the exclusion of pseudo-RH, the investigators will assess if the dosing of medications is correct and drug combination is not suboptimal using the Malawi Standard Treatment Guidelines. Additionally, adherence to medications will be assessed

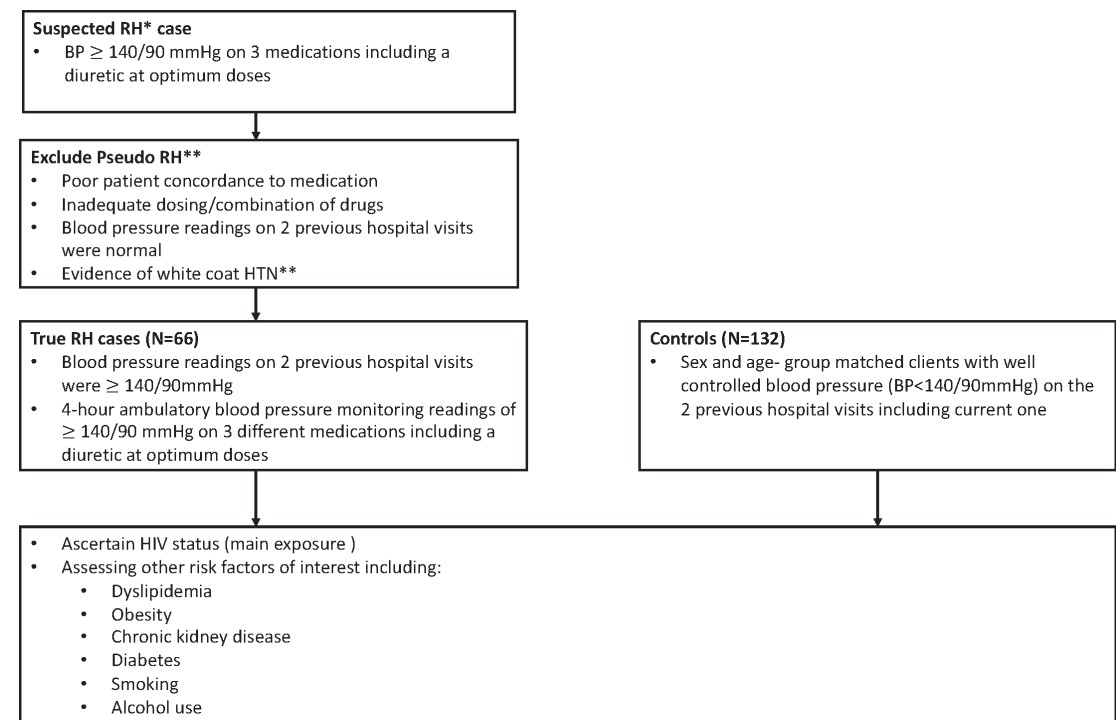

**Figure 1** Description of study design and profile. BP, blood pressure; RH, resistant hypertension.

through the Medication Adherence Scale which has been validated the African cohorts.[13][14] Thereafter, those with correct doses and optimum drug combinations, who are adherent to their medication will be assessed for white-coat hypertension (or lack thereof) by taking blood pressure measurements continuously over 4 hours using an ABPM. In this 4 hour period, the ABPM machine will continuously monitor blood pressure, but it will automatically record blood pressure every 30 min. If a patient still has a blood pressure reading of ≥140/90 mm Hg after this time, they will be defined as having RH and enrolled into the study as a case.

The controls will be sex and age-group-matched individuals who have well-controlled blood pressure (BP<140/90 mm Hg). After obtaining consent in both groups, HIV status will be ascertained. Other factors such as chronic kidney disease, obesity, hypercholesteraemia, diabetes, smoking, alcohol use, ART regimen and duration will be assessed through the collection of medical history, review of notes in health passport books and blood tests.

### Inclusion criteria for cases
► Non-pregnant adults (≥18 years)
► Patients taking at least 3 classes of antihypertensive medication including a diuretic for at least a month.
► Patients with high blood pressure readings of ≥140/90 mm Hg on the last two hospital visits.
► Patients with a blood pressure reading of ≥140/90 mm Hg on the date of enrollment.

### Exclusion criteria for cases
► Patients over 18 years with blood pressure readings of ≥140/90 mm Hg who are taking no more than two classes of antihypertensive medication.
► Patients taking at least three classes of antihypertensive medication but not including a diuretic on their drug regimen.

### Inclusion criteria for controls
► Non-pregnant adults (≥18 years) who have well-controlled blood pressure (BP<140/90 mm Hg) taking any classes of antihypertensive medication.
► Patients must have had well-controlled blood pressure (BP<140/90 mm Hg) on their last two hospital visits.

### Exclusion criteria for controls
► Patients over 18 years with blood pressure readings of ≥140/90 mm Hg.
► Patients over 18 years with well-controlled blood pressure (BP<140/90 mm Hg), who are not taking any antihypertensive medication.

### Data collection
Details of the data that will be collected are outlined in table 1. The research staff will first collect demographic data from all participants. Thereafter, participants will be asked about their medical and social history. For this study, information on previous diagnoses of hypertension, HIV, kidney disease, diabetes and high cholesterol will be collected. Information including duration of illness and dose of medication will also be sought. Finally, (BMI) will be calculated and categorised by age, sex and percentile.

**Table 1** Details of data to be collected

| | Enrolment |
|---|---|
| Baseline characteristics | |
| Name | X |
| Age | X |
| Sex | X |
| Phone number | X |
| Social history | |
| Alcohol intake | X |
| Smoking | X |
| Medical history | |
| HIV | X |
| Hypertension | X |
| Kidney disease | X |
| Hypercholesteraemia | X |
| Diabetes | X |
| Other medical diagnoses | X |
| Anthropometric measurements | |
| Height | X |
| Weight | X |
| Blood pressure measurement | |
| Systolic blood pressure | X |
| Diastolic blood pressure | X |
| 4 hour ambulatory blood pressure monitoring | X |
| Blood results | |
| Creatinine | X |
| Total cholesterol | X |
| HIV rapid test | X |

### Objective 1: Assessing the association between HIV infection and RH

In a private environment, participants will first be asked if they are aware of their HIV status. HIV status will be deemed positive if the individual is on ART or has a documented positive result in their health passport. Those who have been taking ART will provide more information on regimen(s) and how long they have been taking the drugs. This includes asking if they have been switched on ART previously.

Individuals with documented negative HIV test results in their health passports within the last 3 months will be marked as negative. Three months have been employed in tandem with the window period for HIV infection. Rapid antibody tests (which is what will be used in this study) can accurately detect HIV infection 90 days after exposure. An HIV test following the testing algorithm that is stipulated in the Malawi HIV Clinical Guidelines[15] will be offered after relevant informed consent and pretest counselling to those who are uncertain about their status or

who verbally indicate that they have a negative result without documentation in their health passport (but are willing to have their HIV status confirmed). Post-test counselling will be offered, and if found positive, participants will be referred for ART initiation. The HIV test will be done by trained providers at the Accidents and Emergency department at QECH.

### Objective 2: Identifying other independent risk factors or predictors of RH

In previous studies,[7 8 16] it has been shown that clinical conditions including chronic kidney disease, obesity and dyslipidaemia are associated with a higher risk of RH. In this study, these three conditions will be studied as (modifiable) risk factors. Screening for the presence of these comorbidities is part of the standard of care in patients in hypertensive patients. In circumstances where recent (<3 months) blood results to monitor for these conditions are not available, blood samples will be collected to conduct two biochemistry tests: creatinine and total cholesterol from both cases and controls. The creatinine results will be used to calculate the estimated Glomerular Filtration Rate that will determine the presence of chronic kidney disease; 5 mLs of blood will be drawn in a red top test tube to conduct the aforementioned tests. These tests will be conducted at the Malawi-Liverpool-Wellcome Research Programme's laboratory. Additionally, the weight and height of clients will be measured to calculate the BMI. Furthermore, information on diabetes, smoking, alcohol use, ART regimen and duration (for those living with HIV) will be collected through participant interviews and from hospital records.

### Primary outcome

The primary outcome of the study is the adjusted OR, with a 95% CI, of the association between RH and HIV. This will be defined as the odds of HIV in RH patients divided by the odds of HIV in hypertensive patients without RH.

### Secondary outcomes

The secondary outcome measures are the ORs, with 95% CIs, of the independent association between the covariates representing other potential risk factors: cholesterol, obesity, kidney disease and RH.

### Sample size

In calculating the sample size, a threefold relative difference in the odds of HIV infection in cases compared with controls was considered clinically significant. Assuming an HIV prevalence of 10.6% among controls, as is seen in the general population,[10] a sample size of 66 cases and 132 controls (a total of 198) will provide at least 80% power, at a 5% level of significance, to detect a threefold relative difference in the odds of exposure (HIV infection) between those with- and without RH (ie, the OR of HIV infection in cases relative to controls=3). These computations were conducted using STATA V.17, using the function of estimating sample size for a matched case–control study.

## Data management

All data in the study will be collected and managed using the Open-Data Kit software platform. Data will be entered by the study team, and the data accuracy will be verified by the study principal investigator. Only study team members will have access to protected health information. All study-related information will be stored securely at the study site. Tablet computers used to collect the data will be password protected. Any physical participant information will be stored in locked file cabinets in areas with limited access. Patients will not be identified by name in any reports on this study. The study PI and coinvestigators will have access to the final dataset of the study.

## Data analysis plan

The number and percentages of cases and controls identified, and those recruited into the study and participants included in the final analysis will be reported. Additionally, the median (range) of baseline continuous variables (age, weight, height, BMI, blood pressure, total cholesterol and creatinine values in both groups) will be described for both cases and controls. Furthermore, the prevalence of other risk factors, HIV, obesity, hypercholesteraemia and chronic kidney disease in both RH cases and non-RH controls, will be described. In participants living with HIV, different ART regimens and documented viral load and/or CD4 counts, in both controls and cases, will be compared.

To determine the primary and secondary outcomes, two main steps will be carried out. First, the crude ORs of the association between RH and exposure characteristics of interest: HIV, obesity, chronic kidney disease and hypercholesteraemia will be computed in univariate analysis with conditional logistic regression models. Similarly, the relationships between other exposure variables: smoking, alcohol intake and diabetes and RH will be established. In participants living with HIV, we will explore the relationship between the different ART regimens and RH. Point estimates, their CIs and statistical significance, assessed at a 5% level of significance, will be used to determine the strength of the association. Except for HIV, only covariates that show narrow CIs in this univariate analysis will be carried on to the next step.

Second, using a backward stepwise approach in a conditional logistic regression model, the adjusted association between RH and HIV will be ascertained. Covariates to be adjusted for will be those that show a narrow CI for the crude OR in the first step above. For all analyses, the $\chi^2$ test will be used to compare categorical data between cases and controls, while the Wilcoxon Rank Sum test will be used for continuous data. Reporting of results will be in line with the Strengthening the Reporting of Observational Studies in Epidemiology guidelines.[17 18] All analyses will be conducted using STATA V.17 software. In a sensitivity analysis, directed acyclic graphs (DAG), as previously described,[19 20] will be used to evaluate potential predictors of RH or confounders of the association between RH and HIV. Where there is a discrepancy between the identified confounders in the DAGs and the backward stepwise method, covariates identified using DAGs will be carried forward. The use of the two methods in determining confounders or other independent predictors to include in the final model has been opted to ensure robustness in making causal inferences in the association between RH and HIV.

## Patient and public involvement statement

Patients and the public were not involved in setting the research priorities, defining the research question or determining the outcomes of this study.

## ETHICS AND DISSEMINATION

This study has been approved by the College of Medicine Research Ethics Committee (COMREC) with approval number P.05/22/3637. Written consent will be sought from participants before enrolment into the study. Findings from this study will be disseminated through submission for publication in an international, open-access, peer-reviewed journal. Anonymised data from the study will be made available on request from the authors.

**Contributors** JG and CGB conceptualised the research question. JG drafted the protocol manuscript. MN, FK and CGB reviewed the protocol manuscript. JG and MN will have an oversight of data collection with supervision from FK and CGB.

**Funding** JG is supported by an institutional training grant awarded as part of the Wellcome Strategic award number 206545/Z/17/Z to The Malawi-Liverpool-Wellcome Trust Clinical Research Programme (MLW), administered under the joint MLW/Kamuzu University of Health Sciences Training CommitteeCGB is supported by Wellcome and the National Institute for Health and Care Research (NIHR) under the Wellcome International Training Fellowship scheme, award number 222011/Z/20/Z.

**Competing interests** None declared.

**Patient and public involvement** Patients and/or the public were not involved in the design, or conduct, or reporting or dissemination plans of this research.

**Patient consent for publication** Not required.

**Provenance and peer review** Not commissioned; externally peer reviewed.

**ORCID iD**
Clifford George Banda http://orcid.org/0000-0002-0757-5259

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
