## [Reviewer comments · BMJ Open]

ARTICLE DETAILS

TITLE (PROVISIONAL)	Association between HIV and treatment-resistant hypertension in Malawian adults: a protocol for a case-control study
AUTHORS	Gondwe, Josephine; Ndovie, Maclean; Khuluza, Felix; Banda, Clifford

VERSION 1 – REVIEW

REVIEWER	Alana Brennan Boston University, Global Health
REVIEW RETURNED	02-Feb-2023

GENERAL COMMENTS	Major concerns: 1. There is no mention of the time period of this study? What are the anticipated dates of data collection/patient enrollment?2. Has the data already been collected? I identified the following link - http://rscarchive.kuhes.ac.mw/handle/20.500.12988/1108, if so, then it fails to meet BMJs publication requirements for publishing study protocols.3. Authors state, “The global burden of RH has been estimated to be around 94 10.3% (3), and previous studies have estimated the burden of RH in Africa to be 95 ~12% (8).” Authors claim that the outcome is rare and justifies a case-control study. If we use the 10% cutoff for a rare outcome, that is not the case. Do the authors have any estimates from previous literature on the prevalence of RH in HIV populations?4. Control selection: Authors state, “Controls will be defined as hypertensive patients with well-controlled blood pressure (BP <140/90mmHg) regardless of how many antihypertensive medications they are currently taking.” Why are authors allowing the controls to be on any type of hypertension medication? Can they justify why they would allow this in the controls?5. Authors state on line 204, “Thereafter, blood pressure will be measured using ambulatory blood pressure monitoring (ABPM) cuffs over 4 hours. The 4-hour ABPM will be adopted in this study since previous studies have shown that a 4-hour ABPM window can be used to accurately represent daytime blood pressure (12).” I would move this up and define it more clearly earlier around line 127 when you define cases.6. State ABPM at first use of the term on line 127.7. Outcome definition is a bit unclear. What is the frequency of the BP measurements over 4-hour period when they are measuring ABPM? Is it every 30 minutes?8. Why is time on HTN medication not considered? Isn't there a specific period of time that the HTN drugs need to work when a patient is initially put on them? Weeks to months, depending? If they are not requiring enough time for the drugs to take effect then wouldn't there be a chance that they would document someone as
---

	have RH when they don't if they are early on in their treatment for HTN? 9. Has the Medication Adherence Scale been validated in an African cohort? 10. Was the 4-hour ABPM validated in an African cohort? You cite - Chanudet Xavier, Chau Phong Nguyen, Larroque Pierre. Short-term representatives of daytime and night-time ambulatory blood pressures. J Hypertens. 1992 Jun;10(6):595–600. I can't access the article to determine the study population. 11. I think the authors have 1 objective as stated in objective #2 - Identifying risk factors or predictors of RH, of which HIV is one of those factors they are evaluating. I don't think it is necessary to state 2 objectives as their regression models will have all variables in them, including HIV. 12. Authors state in their sample size section, "In calculating the sample size, a three-fold relative difference in the odds of HIV infection in cases compared with controls was considered clinically significant." How to they estimate that there would be a 3-fold difference? Was that stated in the literature previously? You know that RH is roughly 10-12% in HIV-uninfected, however where do you get the estimates for the population of PLWH? If the difference is much smaller than 3-fold you will need a much larger sample size to assess. 13. In the data analysis plan section, the authors focus on statistical significance in variables and propose using backwards deletion to determine their final set of predictors. I highly discourage this as it is a case control study of a rare outcome, and they have a very small sample size. If they base the results on significance, they may overlook important associations between predictors and the outcomes because their sample size is too small. I would recommend focusing on the precision of your estimates (confidence intervals) and the association of the predictors and the outcome of interest, not a p-value. Are the authors familiar with directed acyclic graphs (DAGs)? This would be a better way to represent the relationship between potential predictors, confounders and effect measure modifiers in the relationship between HIV and RH. I would highly discourage focusing on significance. 14. I assume the x2 test will be presented in table 1. Please keep any p-values out of table 1. You can refer to the STROBE statement on the best way to present the data. https://www.strobe-statement.org/. It is good practice to follow the STROBE statement when presenting work.
--	---

REVIEWER	Martin Muddu Makerere Univ
REVIEW RETURNED	11-Feb-2023

GENERAL COMMENTS	Comments to the authors The authors are answering an important research questions given the high burden of hypertension among adult PLHIV and the need for treatment options. This is a well written protocol manuscript. However, the authors need to address the following comments to improve it. Most importantly, the issue of potential selection bias needs special attention. In line 43, authors mention that the prevalence of HIV will be ascertained in both groups (cases and controls). However, for a case control study, it would be better to determine the association of HIV with RH using multivariate analysis. Please dress this issue. Lines 94-96, when you mention the prevalence of RH as 10.3% and
--

	12%. What are the denominators? In line 132, you mention a 1:2 ratio of cases: control. Please provide justification for that strategy. The sampling strategy is inadequately described. It is not clear how the investigator will minimize selection bias among the cases and controls. For example, selection of many HIV positive patients among the cases may imply that HIV is a risk factor for RH yet it may not be the case and this will lead to reporting of wrong results. In line 220, you mention a rapid antibody test for HIV. However, the WHO recommends a testing algorithm for HIV. What algorithm will you use for HIV testing? In line 232, you mention chronic kidney disease as a factor that you will assess. How will you determine if a patient has chronic kidney disease? In line 258, you mention the prevalence of HIV of 10.6%. Is this a prevalence for all age groups or is it for those aged 18 years and above, your target population? If this is a population prevalence, then the sample size should be calculated based on the prevalence of HIV in persons aged 18 years and above.
--	---

VERSION 1 – AUTHOR RESPONSE

Reviewer: 1

Major concerns:

1. There is no mention of the time period of this study? What are the anticipated dates of data collection/patient enrollment?

Response: The period for the study is from January 2022 to August 2023. This has now been included in the protocol manuscript I in lines 146-148. We are currently recruiting participants and anticipate completion in June 2023.

2. Has the data already been collected? I identified the following link - <http://rscarchive.kuhes.ac.mw/handle/20.500.12988/1108>, if so, then it fails to meet BMJs publication requirements for publishing study protocols.

Response: At the time of submission in October 2022, data collection had not yet begun. Participant recruitment began in February 2023 and is ongoing.
The above-cited link repository is for the planned research within the Kamuzu University of Health Sciences' research database.

3. Authors state, "The global burden of RH has been estimated to be around 94 10.3% (3), and previous studies have estimated the burden of RH in Africa to be 95 ~12% (8)." Authors claim that the outcome is rare and justifies a case-control study. If we use the 10% cutoff for a rare outcome, that is not the case. Do the authors have any estimates from previous literature on the prevalence of RH in HIV populations?

Response: The prevalence of resistant hypertension in HIV populations is yet to be studied. The estimates of the overall prevalence of RH available in the African cohort include studies from only 5 countries with 4,063 participants. Whilst the 12% was a pooled prevalence, the individual study's prevalence themselves varied widely from 4.9% to 19%. This heterogeneity shows the uncertainty in classifying RH as a common disease. Additionally, resistant hypertension has not been studied in Malawi, thus this case-control study design is most appropriate to conduct in our setting for both hypothesis generation and testing.

4. Control selection: Authors state, "Controls will be defined as hypertensive patients with well-controlled blood pressure (BP <140/90mmHg) regardless of how many antihypertensive medications they are currently taking." Why are authors allowing the controls to be on any type of hypertension medication? Can they justify why they would allow this in the controls?

Response: Resistant hypertension is defined as uncontrolled blood pressure despite treatment with at least 3 medications of different classes (including diuretics) at optimal doses. The difference we aim to detect is in comparison to hypertensive individuals whose blood pressure is well-controlled. This would, in practice, mean that the number of antihypertensive medications to achieve BP control would vary in the control. We will then ascertain the exposure status in both groups to see if HIV is associated with uncontrolled blood pressure.

5. Authors state on line 204, "Thereafter, blood pressure will be measured using ambulatory blood pressure monitoring (ABPM) cuffs over 4 hours. The 4-hour ABPM will be adopted in this study since previous studies have shown that a 4-hour ABPM window can be used to accurately represent daytime blood pressure (12)." I would move this up and define it more clearly earlier around line 127 when you define cases.

Response: Many thanks. That has been amended and defined in lines 138 to 140.

6. State ABPM at first use of the term on line 127.

Response: This has been amended in now line 137.

7. Outcome definition is a bit unclear. What is the frequency of the BP measurements over the 4-hour period when they are measuring ABPM? Is it every 30 minutes?

Response: The ABPM machine is designed to continuously monitor blood pressure over a period, but it will automatically record blood pressure every 30 minutes. Line 179-181 now includes this information.

8. Why is time on HTN medication not considered? Isn't there a specific period of time that the HTN drugs need to work when a patient is initially put on them? Weeks to months, depending? If they are not requiring enough time for the drugs to take effect then wouldn't there be a chance that they would document someone as have RH when they don't if they are early on in their treatment for HTN?

Response: Most classes of antihypertensive drugs reach steady-state and typically start working within two days of taking the first dose, but this can indeed take up to 2-4 weeks for the patient to realize their full effect. Line 198-199 now clearly stipulates that patients must have been on the medications for at least a month. As resistant hypertension is defined as uncontrolled blood pressure despite using at least 3 medications, it is likely that for a patient to get 3 classes of drugs they have been on antihypertensive medications for at least a month. Normally, patients are started on a single agent and the doses are gradually titrated up before the addition of a second or third agent. Additionally, line 200 specifies that cases should have had high blood pressure readings in their last 2 visits (which are typically 2 months apart) to ascertain that this high blood pressure is indeed persistent and thus avoid misclassification.

9. Has the Medication Adherence Scale been validated in an African cohort?

Response: This scale was validated in the African cohort and the reference has been added in lines 175-176.

10. Was the 4-hour ABPM validated in an African cohort? You cite - Chanudet Xavier, Chau Phong Nguyen, Larroque Pierre. Short-term representatives of daytime and night-time ambulatory blood pressures. *J Hypertens*. 1992 Jun;10(6):595–600. I can't access the article to determine the study population.

Response: The referenced study was conducted in France. There has not been a similar study within an African setting, The investigators adopted the 4-hour ABPM monitoring as it is more feasible within our context and in most African settings than the 24-hour monitoring.

11. I think the authors have 1 objective as stated in objective #2 - Identifying risk factors or predictors of RH, of which HIV is one of those factors they are evaluating. I don't think it is necessary to state 2 objectives as their regression models will have all variables in them, including HIV.

Response: In now line 251, clarification has been added that the second objective is aiming to identify other independent risk factors of RH apart from HIV

12. Authors state in their sample size section, "In calculating the sample size, a three-fold relative difference in the odds of HIV infection in cases compared with controls was considered clinically significant." How do they estimate that there would be a 3-fold difference? Was that stated in the literature previously? You know that RH is roughly 10-12% in HIV-uninfected, however, where do you get the estimates for the population of PLWH? If the difference is much smaller than 3-fold you will need a much larger sample size to assess.

Response: This estimate was deemed clinically relevant as there is no previous literature on the prevalence of resistant hypertension in people living with HIV to base the comparison with HIV-negative patients. What is known is the prevalence of hypertension among people living with HIV is about 34.7% (reference number 3).

13. In the data analysis plan section, the authors focus on statistical significance in variables and propose using backward deletion to determine their final set of predictors. I highly discourage this as it is a case-control study of a rare outcome, and they have a very small sample size. If they base the results on significance, they may overlook important associations between predictors and the outcomes because their sample size is too small. I would recommend focusing on the precision of your estimates (confidence intervals) and the association of the predictors and the outcome of interest, not a p-value. Are the authors familiar with directed acyclic graphs (DAGs)? This would be a better way to represent the relationship between potential predictors, confounders and effect measure modifiers in the relationship between HIV and RH. I would highly discourage focusing on significance.

Response: This is well noted. In now lines 320-323, we have clarified that "point estimates, their confidence intervals and statistical significance, assessed at a 5% level of significance, will be used to determine the strength of the association. Except for HIV, only covariates that show narrow confidence intervals in this univariate analysis will be carried on to the next step"

Additionally, we have incorporated the use of DAGs as a sensitivity analysis as indicated in lines 332-341 as follows: "In a sensitivity analysis, directed acyclic graphs (DAG), as previously described, will be used to evaluate potential predictors of RH or confounders of the association between RH and HIV. Where there is a discrepancy between the identified confounders in the DAGs and the backward stepwise method, covariates identified using DAGs will be carried forward. The use of the two methods in determining confounders or other independent predictors to include in the final model has been opted to ensure robustness in making causal inferences in the association between RH and HIV".

14. I assume the χ^2 test will be presented in table 1. Please keep any p-values out of table 1. You can refer to the STROBE statement on the best way to present the data. <https://www.strobe-statement.org/>. It is good practice to follow the STROBE statement when presenting work.

Response: In lines 330-331, we have highlighted that all reporting will be in line with STROBE guidelines. This will include reporting of findings in Table 1.

Reviewer: 2

Comments to the Author:

The authors are answering an important research questions given the high burden of hypertension among adult PLHIV and the need for treatment options. This is a well written protocol manuscript. However, the authors need to address the following comments to improve it. Most importantly, the issue of potential selection bias needs special attention.

The controls in the study will come from the same population giving rise to the cases. In our study, both cases and controls will come from the same clinic. Furthermore, in the controls will be identified through simple random sampling and not convenient sampling. In both cases and controls, ascertainment of HIV status will occur after recruitment into the study. This will minimise any selection bias. Further details are under response to comment 4 below.

1. In line 43, authors mention that the prevalence of HIV will be ascertained in both groups (cases and controls). However, for a case control study, it would be better to determine the association of HIV with RH using multivariate analysis. Please adress this issue.

Response: Line 38-40 describes that a multivariable conditional logistic regression will be used to ascertain the association. The statement in line 33 meant that in both cases and controls, we will ascertain the percentage of HIV, and this will be shown in table 1. The statement (now lines 33-34) has been changed to properly explain that HIV status will be ascertained in both groups (cases and controls)

2. Lines 94-96, when you mention the prevalence of RH as 10.3% and 12%. What are the denominators?

Response: The global burden estimate of 10.3% is from a meta-analysis of 63,554 participants from 49 studies. The estimates of ~12% in Africa is from a systematic review of 4,068 participants from 5 countries.

3. In line 132, you mention a 1:2 ratio of cases: control. Please provide justification for that strategy.

Response: The 1:2 ratio was chosen to increase statistical power in detecting a difference between the cases and controls while bearing in mind that there would be less number of cases

4. The sampling strategy is inadequately described. It is not clear how the investigator will minimize selection bias among the cases and controls. For example, selection of many HIV positive patients among the cases may imply that HIV is a risk factor for RH yet it may not be the case and this will lead to reporting of wrong results.

Response: There should not be a concern for selection bias in the choice of cases since every potential case of RH will be screened regardless of HIV status. The HIV status of each case will be determined after a patient has been classified as a case. The concern for selection bias would be more prominent in the selection of controls. To minimise this:

- a. The controls in the study will come from the same population giving rise to the cases.
- b. The selection of controls will be by simple random sampling technique of all patients presenting at the clinic on the day a case has been identified.
- c. Ascertainment of HIV status would only happen after a control has already been selected and recruited into the study.

Additionally, no recruitment of participants will occur at the antiretroviral therapy clinic. The above explanation has been highlighted in now lines 150-157.

Response:

5. In line 220, you mention a rapid antibody test for HIV. However, the WHO recommends a testing algorithm for HIV. What algorithm will you use for HIV testing?

Response: HIV testing will follow the algorithm stipulated in the Malawi Integrated Guidelines and Standard Operating Procedures for the clinical management of HIV in children and adults (2022). The reference has been added in line 242-243.

6. In line 232, you mention chronic kidney disease as a factor that you will assess. How will you determine if a patient has chronic kidney disease?

Response: As we are collecting blood samples for creatinine, we will use the creatinine results to calculate the estimated glomerular filtration rate (eGFR) and determine CKD from the results. This has been reflected in the protocol on line 259 – 261.

7. In line 258, you mention the prevalence of HIV of 10.6%. Is this a prevalence for all age groups or is it for those aged 18 years and above, your target population? If this is a population prevalence, then the sample size should be calculated based on the prevalence of HIV in persons aged 18 years and above.

Response: The prevalence of 10.6% has been calculated from a population aged 15 and older however the 3-year difference (15 to 18 years) would not be significant enough to distort the sample size calculations.

VERSION 2 – REVIEW

REVIEWER	Martin Muddu Makerere Univ
REVIEW RETURNED	10-Apr-2023
GENERAL COMMENTS	The authors have addressed the comments that I raised.